# Treatment Options in AF Patients with Cancer; Focus on Catheter Ablation

**DOI:** 10.3390/jcm11154452

**Published:** 2022-07-30

**Authors:** Silvia Garibaldi, Michela Chianca, Iacopo Fabiani, Michele Emdin, Marcello Piacenti, Claudio Passino, Alberto Aimo, Antonella Fedele, Carlo Maria Cipolla, Daniela Maria Cardinale

**Affiliations:** 1Cardiology Division, Fondazione Toscana Gabriele Monasterio, 56124 Pisa, Italy; sgaribaldi@ftgm.it (S.G.); emdin@ftgm.it (M.E.); piacenti@ftgm.it (M.P.); passino@ftgm.it (C.P.); 2Health Science Interdisciplinary Center, Scuola Superiore Sant’Anna, 56127 Pisa, Italy; michela.chianca@santannapisa.it (M.C.); alberto.aimo@santannapisa.it (A.A.); 3Cardioncology Unit, Cardioncology and Second Opinion Division, European Institute of Oncology, Istituto di Ricovero e Cura a Carattere Scientifico, 20141 Milan, Italy; antofedele2010@gmail.com (A.F.); carlo.cipolla@ieo.it (C.M.C.); daniela.cardinale@ieo.it (D.M.C.)

**Keywords:** cancer, atrial fibrillation, pulmonary vein isolation ablation, anticoagulation, cardio-oncology

## Abstract

Longer life expectancy along with advancements in cancer and atrial fibrillation (AF) therapies and treatment strategies have led to an increase in the number of individuals with both diseases. As a result, the complicated management of these patients has become crucial, necessitating individualised treatment that considers the bi-directional relationship between these two diseases. On the one hand, giving appropriate pharmaceutical therapy is exceptionally difficult, considering the recognised thromboembolic risk posed by AF and malignancy, as well as the haemorrhagic risk posed by cancer. The alternative pulmonary vein isolation (PVI) ablation, on the other hand, has been inadequately explored in the cancer patient population; there is yet inadequate data to allow the clinician to unambiguously select patients that can undertake this therapeutic intervention. The goal of this review is to compile the most valuable data and supporting evidence about the characteristics, care, and therapy of cancer patients with AF. Specifically, we will evaluate the pharmaceutical options for a proper anticoagulant therapy, as well as the feasibility and safety of PVI in this population.

## 1. Introduction

Atrial fibrillation (AF), the most frequent persistent cardiac arrhythmia and its prevalence, is rising globally [1]. Moreover, a substantial percentage of cancer survivors experience symptomatic AF as a result of the effectiveness of cancer treatments. Several studies [2,3] have found a link between malignancy and AF; cancer and AF appear to have a bidirectional relationship, presumably due to common risk factors. While early reports showed that cancer patients had a higher incidence of AF after getting medical or surgical cancer treatment, more current data suggest that cancer patients have an increased prevalence of AF even before receiving specific cancer treatment [4]. Individuals with new-onset AF, on the other hand, have a much higher probability of being diagnosed with cancer, particularly in the first three months following diagnosis [5,6]. Although AF and cancer are both characterised by inflammation, the underlying processes that explain the link between these two diseases remain unknown [7].

To date, minimal study has been done on the management of AF in cancer patients [1], and there are still many grey areas in the field of therapeutic management of cancer patients with AF that will require more investigations. In this regard, current guidelines do not offer precise recommendations on the best thrombo-prophylaxis approach for these patients (Figure 1).

Furthermore, as cancer treatments improve and patients live longer after diagnosis, the proportion of people with AF and cancer is estimated to climb even further, with a large portion of this population requiring interventional therapy for symptomatic AF. Hypercoagulable status or circumstances that favor spontaneous bleeding, on the other hand, may raise the risk of peri-procedural thromboembolic or bleeding complications.

This review’s goal is to provide an overview of the most recent research on the management of AF and cancer, with focus on anticoagulant therapy and interventional treatment in this setting.

## 2. Epidemiology

The prevalence of AF in cancer patients is approximately 20% [5,8,9,10] and it was found to be independent of the type of malignancy [10,11]. The first three months following a cancer diagnosis were associated with the highest risk of developing new AF [5,11], which gradually decreased after six months, although continued to be higher in cancer patients for up to five years [11]. Therefore, patients with new-onset AF have a considerably greater risk of cancer diagnosis, especially in the first three months after diagnosis [5,6].

## 3. Anticoagulant Therapy in Cancer Patients

Emerging evidence suggests that cancer is associated with increased thromboembolic and bleeding risks, making anticoagulation management challenging in cancer patients for any indication. For several decades, vitamin K antagonists (VKAs) have been used to minimize the risk of stroke and systemic embolism in individuals with AF. In individuals with non-valvular AF, direct oral anticoagulants (DOACs) are now indicated as the first-line anticoagulant treatment and are currently increasingly being prescribed; because of the direct interaction of cancer with the coagulation system and the influence of chemotherapy, cancer patients are at a significant risk of both thromboembolic and bleeding events [4]. Antithrombotic therapy in cancer patients with AF is poorly supported by clinically relevant evidence, and just one position paper has been written on the subject [12]. Therapeutic low-molecular-weight heparin (LMWH), a vitamin K antagonist (VKA; for example, warfarin), or a non-VKA oral anticoagulant (NOAC) are all anticoagulation choices if the international normalised ratio control is stable and successful. In cancer patients with metastatic cancer and a high risk of bleeding, warfarin is frequently avoided due to the possibility of variations in the international normalised ratio, with LMWH typically considered to be the preferred choice. Drug interactions, malnutrition, vomiting, and liver dysfunction can all cause erratic bioavailability and anticoagulation levels. Conversely, LMWHs have predictable pharmacokinetic profiles and very few drug interactions [13].

Due to a lack of data, the role and safety of NOACs in this patient population are yet unknown.

Several post-hoc analyses of RCTs or observational studies have examined the use of NOACs in comparison to warfarin in AF patients with a history of malignancy, even though no head-to-head RCTs have been conducted for this population (Table 1) [14,15,16,17,18,19,20]. In particular, ROCKET AF [13], ENGAGE AF- TIMI 48 [16], and ARISTOTLE [18] trials reported the effects of NOACs versus warfarin in AF patients with and without cancer. When the data from these three trials were combined, it was shown that patients with and without cancer experienced the same rates of all efficacy and safety outcomes (NOACs versus warfarin) [21].

According to several pieces of research, cancer patients taking NOACs had comparable rates of stroke and bleeding risks to those taking warfarin but had a decreased risk of venous thromboembolism (VTE) [14,15,16,17,18].

In contrast, Kim et al. [17] found that patients on NOACs had reduced rates of thromboembolic and bleeding events, as well as all-cause mortality than those taking warfarin. Russo et al. [25] in a recent comprehensive analysis that included six studies [18,19,20,21], concluded that NOACs appear to be at least as safe and efficacious as conventional anticoagulant therapy using VKAs. Yuqing Deng et al. [21] conducted a more recent meta-analysis in which they compared the effects of NOACs versus warfarin in AF patients and cancer. Using the data of five included studies [14,15,16,17,18], this meta-analysis found that: (1) the risks of SSE, major bleeding, or death did not significantly correlate with cancer status; (2) NOACs had comparable or lower rates of thromboembolic and bleeding events as compared to warfarin, as well as a decreased risk of VTE; and (3) the rates of efficacy and safety outcomes were comparable in AF patients with and without malignancy.

## 4. Left Appendage Closure in Cancer Patients

AF patients with cancer reportedly have a higher bleeding risk with a similar or higher stroke risk than those without cancer [26]. Clinical practice guidelines state that percutaneous left atrial appendage closure (LAAC) is an effective and reliable treatment option for patients with a high embolic risk and a long-term anticoagulation contraindication [4]. However, scarce data are available on the procedural complication risk and outcomes of LAAC among cancer patients.

In a retrospectives study, Samuel A Shabtaie [27] demonstrated a feasibility of LAAC in patients with cancer getting a reasonable reduction in stroke and bleeding risk.

Toshiaki Isogai et al. in a recent retrospective analysis showed no significant difference in in-hospital mortality in patients treated with LAAC, regardless of cancer status (active or history).

On the other hand, the in-hospital ischemic stroke/TIA rate was significantly higher in the active-cancer group, but not in patients with a history of cancer or no-cancer groups.

Furthermore, the active-cancer group did not experience an early readmission for ischemic stroke/TIA, indicating that LAAC is beneficial in cancer patients in the short term [28].

In conclusion, percutaneous left atrial appendage closure, according to the expert team, could be a possibility for patients with cancer and nonvalvular AF who are ineligible for anticoagulant medication, and who have a life expectancy of more than a year, [29].

## 5. Pulmonary Vein Isolation (PVI) Ablation

The discovery of an AF trigger from within the pulmonary veins by Hassaguerre and colleagues in 1998 was a watershed moment in the field of cardiac electrophysiology, revealing a novel mechanism behind AF onset as well as a new prospective target therapy [30]. Cardiovascular ablation to isolate the pulmonary veins has advanced over the past 20 years, starting with the first described segmental ostial pulmonary vein ablation [31] and progressing to the use of 3D electro-anatomical mapping to guide ablation [32] and, finally, to the antral isolation of pulmonary veins, which has been shown to be more effective and safe than ostial segmental isolation of each individual pulmonary vein [33]. Pulmonary vein antral isolation is currently recommended for all AF ablation treatments (class I indication, level of proof A) [34]. A variety of energy modalities have been used for catheter ablation of AF, including radiofrequency (RF) and cryoablation (CB), which are the two most used sources in clinical practice. The most popular technique is the application of radiofrequency current in a point-by-point mode, which causes tissue heating and results in cellular necrosis; the alternative technique is the implementation of cryogenic energy with a balloon in a single-step mode, which results in necrosis through freezing. Fluoroscopy is only seldom necessary for radiofrequency ablation of atrial fibrillation because catheter guidance is accomplished by using an electro-anatomical mapping device [35]. Cryoablation for atrial fibrillation, on the other hand, necessitates more thorough fluoroscopic guidance to place the balloon catheter at the pulmonary veins and ensure optimal tissue freezing [36].

Three significant prospective RCTs have so far compared cryoballoon ablation with RF in patients with AF and found equal efficacy and safety between the two techniques [37,38,39].

### 5.1. Radiation Exposure during Atrial Fibrillation Catheter Ablation

One of the major risks related to PVI ablation is radiation exposure [40].

The effects of ionizing radiation exposure are both deterministic and stochastic. The latter is especially important in young patients with increased radiosensitivity and a longer life expectancy, as well as in patients who undergoing a large cumulative radiation dose for long, difficult, or recurrent treatments [41,42,43,44,45]. The radiation exposure, on the other hand, could result in a large cumulative dosage and a lifelong radiation risk for the electrophysiological staff [46,47]. As a result, the European Directives and the International Commission on Radiological Protection urge that physicians utilize diagnostic reference values to guide their radiation use [48].

Therefore, it is crucial to reduce the amount of ionising radiation that patients and personnel are exposed to. Few studies have used state-of-the-art methods to estimate the cancer risk associated with radiation exposure during cardiac diagnostic tests and therapeutic procedures. In comparison to other ablation techniques, catheter ablation for the treatment of atrial fibrillation is complex and necessitates longer procedural and fluoroscopy periods, subjecting patients, doctors, and nurses to more radiation. A study by Macle et al. showed that the median patient exposure was 0.0011 Gy for AF ablation compared with 0.0005 Gy for common flutter and 0.00056 Gy for accessory pathway ablation (*p* < 0.01) [49].

Lickfett et al. in their study showed that the additional lifetime risk for a fatal malignancy radiation exposure related with ablation of AF was 0.15% for female patients and 0.21% for male patients. However, catheter ablation was performed without electroanatomic mapping systems and kerma-area product meters were unavailable [50].

In recent years, the development and widespread use of electro-anatomical mapping systems in conjunction with the transesophageal or intracardiac echocardiography during PVI ablation with RF has resulted in a significant reduction in ionizing radiation exposure, leading to a decreased risk of cancer incidence and mortality [51,52,53,54,55,56,57,58]. Recently, for the first time, the zero-fluoroscopic approach was described in consecutive patients scheduled for AF ablation, eliminating the potential risk of cancer that are procedure-related [59].

Cryo-energy ablation does not benefit from electro-anatomic mapping systems. Despite a previous paper [39], Russo et al. in a recent retrospective analysis evaluated fluoroscopy exposure data with both ablation techniques observing no significant difference in fluoroscopy time [60].

### 5.2. Catheter Ablation of Atrial Fibrillation in Cancer Patients

New generation catheters and improved ablation methods have made ablation procedures simpler and have extended their use to more complex cases [61]. Given that ablation is not contraindicated by cancer prognosis, cardio-onco-hematology teams may explore ablation in highly chosen individuals when alternative heart rate or rhythm management measures have failed or when there is a significant likelihood of interactions with cancer treatments [62]. Only a few studies have analyzed PVI in cancer patients (Table 2); Arun Kanmanthareddy et al. found that the catheter ablation of AF with radiofrequency was secure and reliable in 15 patients with a history of prior pneumonectomy, 10 of which were for lung cancer [63].

Similarly, Shabtaie et al. [64] demonstrated the viability of performing atrial fibrillation ablation procedures (23% of arrythmias) in patients with neuroendocrine tumors, including those with metastatic disease, carcinoid syndrome, and carcinoid heart disease. Despite promising findings on the use of PVI in cancer patients, these studies have some limitations: Firstly, the sample of patients analysed is rather limited, with only a few tumour types being analysed. Secondly, no information is provided on the management of periprocedural anticoagulant therapy. To our knowledge, only two studies evaluate the safety of ablative therapy of atrial fibrillation in patients with history of different kinds of cancer.

According to Giustozzi and coworkers [65] cancer survivors undergoing catheter ablation with radiofrequency for AF had higher periprocedural bleeding rates. A total of 184 patients were enrolled in the trial, with 21 (11%) of them having a history of cancer. The most common cancer sites were gastrointestinal (36%), breast (23%), and genitourinary (18%). At the time of inclusion, they were receiving treatment with either vitamin-K antagonists or novel oral anticoagulants (NOACs). An interrupted periprocedural anticoagulant protocol was applied. VKA was discontinued for 4–5 days and replaced with low-molecular-weight heparin bridging, attaining a target international normalised ratio (INR) of 1.7 at procedure time. Low-molecular-weight heparin bridging was continued after discharge until therapeutic INR (>2) was reached. NOACs were typically shut down for 24 h without bridging and, if possible, resumed the next morning of the surgery. Intravenous heparin was administered during the ablation process following transseptal puncture to maintain the goal activated clotting time of 300 s. Following the procedure, an active clotting time control continuous intravenous heparin infusion was administered till the next morning. Cancer survivors had a higher incidence of clinically significant bleeding at the 1-month follow-up than non-survivors did. The anticoagulation type was not linked to bleedings. On the other hand, Charlotte Eitel et al. [66] evaluated for the first time the safety and efficacy of isolation with the cryoablation of pulmonary veins in patients with a history of cancer to a matched cohort of patients without cancer.

The majority of the participants were cancer survivors (62 of 70, or 88.6%), with the remaining eight having active tumour disease at the time of ablation. Genitourinary cancer accounted for 30% of the malignancies studied, followed by breast cancer (28.6%), haemato-oncological cancer (12.9%), gastrointestinal cancer (11.4%), head and neck cancer (5.7%), and lung cancer (2.9%). Finally, 8.6% of patients had a history of multiple tumour entities. Patients on vitamin K antagonists were given continuous anticoagulation with a two to three target INR. One dosage of direct oral anticoagulants (DOAC) was paused the morning of the surgery and restarted six hours later in patients who were already taking them. Then, utilising a typical ablation technique with a predetermined freeze-cycle duration of 180 s, all patients underwent CB-PVI using a 28 mm second or fourth generation cryoballoon. After transseptal puncture, heparin was given to patients with a goal activated clotting time of more than 300 s. In contrast to Giustozzi et al., this study discovered that patients with a history of malignancy had comparable rates of arrhythmia-free survival without noticeably different periprocedural complications from the controls.

### 5.3. Periprocedural Anticoagulant

To date, a number of studies have examined the utility of different anticoagulation protocols during the periprocedural stage of catheter ablation of atrial fibrillation, but not in a specific setting of cancer patients.

In the last 15 years, oral anticoagulation has advanced substantially, resulting in a decrease in periprocedural thromboembolic and bleeding complications. These findings are mostly attributable to the adoption of continuous anticoagulation techniques, initially with warfarin (VKA), followed by ablation with a periprocedural therapeutic INR, and finally with DOACs, which have largely replaced warfarin in anticoagulation schemes [67,68].

In this regard, a thorough meta-analysis of observational studies involving 27,402 patients [69] found that those who got uninterrupted medication had a lower risk of periprocedural ischemic stroke or TIA, as well as fewer adverse events and bleeding. Similarly, the randomised COMPARE trial [70] assessed the risk of stroke and bleeding during atrial fibrillation catheter ablation in patients receiving different anticoagulation treatments. In this trial, 1584 patients were randomly assigned to one of two arms, continuous versus discontinuous warfarin use. During procedures, the ACT was kept >300 s in both groups.

According to this study, those who stop using warfarin had a lower risk of periprocedural stroke or TIA than those who continue taking it (0.25% vs. 3.7%/1.3%) without seeing an increase in hemorrhagic complications [70]. In the latest years, the use of DOAC in PVI patients has developed quickly, exhibiting effectiveness and safety in comparison to the conventional VKA [71]. Numerous pieces of research have been conducted on the administration of oral anticoagulation during PVI. The feasibility, efficacy, and safety of DOACs in the context of PVI have been highlighted in particular by four RCTs with a comparable study design [72,73,74,75,76]. Randomised control trials [77,78,79], as well as the current European and American Guidelines [34,80], indicate that uninterrupted (or little interrupted) DOACs offer a reliable substitute for a continuous-VKA approach, with a low risk of thromboembolic events and haemorrhage.

There are no indications regarding the management of periprocedural anticoagulant therapy in cancer survivors undergoing atrial fibrillation ablation. Large-scale studies are needed to determine the most effective periprocedural anticoagulation strategy for this particular population.

## 6. Discussion

Since the improvement of cancer therapeutic treatments and a higher life expectancy, the occurrence of coupled AF and cancer has risen over time [12]; cancer can cause atrial fibrillation through a variety of processes, including chemotherapy, potential paraneoplastic manifestations, and inflammation [81]. On the other hand, cancer is also known to promote coagulopathies by providing direct damage to endothelium and causing coagulation abnormalities [12]. As a result, the cancer patient with AF requires treatment that considers the potential effects of the co-existence of these two diseases.

Given the thromboembolic risk associated with both AF and cancer, selecting a suitable anticoagulant therapy for individuals with both diseases is a crucial component of cardio-oncology. Vitamin K antagonists are well-known for causing major side effects in cancer patients, particularly during chemotherapy, such as nausea and vomiting, low food intake, and drug–drug interactions. In comparison to VKAs, DOACs would have a faster onset of action, a shorter half-life, and fewer drug–drug interactions. Furthermore, routine blood tests are not required to establish that the patient is inside the anticoagulation therapeutic window.

Several post-hoc analyses of clinical trials [13,14,15,16,17,18] have suggested that NOACs have favourable benefits independent of cancer history, with patients reporting a better outcome in terms of stroke risk, the number of thromboembolic events, and major bleeding. Despite these positive results, NAOCs have several limits in the cancer patient population: for example, they should not be used in conjunction with anticancer drugs or complementary therapies that are potent inducers or P-glycoprotein inhibitors. Furthermore, the retroactive nature of these analyses introduces issues relating to selection bias, exposure underestimation, and outcomes miscalculation. Looking ahead, more data from randomised studies will be needed to determine whether DAOCS therapy is really more successful and safer in this patient population than vitamin K antagonist therapy.

In the case of intolerance or resistance to anti-arrhythmic medication treatment, catheter ablation has recently been investigated as a possible therapeutic alternative. Actually, the ablation therapy in patients with AF and cancer is not well-defined, and the few available data are conflicting [64,65].

The discrepancy between the two above mentioned studies may be explained by several facts. The higher incidence of clinically significant bleeding in cancer survivor patients in the study by Giustozzi et al. could be due in premise to a different management of periprocedural anticoagulant, in accord with the COMPARE trial (interrupted anticoagulant bridged with low-molecular-weight heparin versus uninterrupted anticoagulation). Furthermore, when compared to PVI with cryoenergy, radiofrequency ablation can result in a longer operation time. As a result, more intra-procedural heparin may be required, thereby increasing the risk of bleeding.

For this reason, these results might imply that continuous or minimally interrupted therapy in the absence of bridging with LMWH should be used as the first choice of management protocol for periprocedural anticoagulant therapy in cancer patients who have an intrinsic increased risk of bleeding. The choice regarding the type of energy for PVI ablation is difficult. The radiofrequency, thanks to the use of a 3-dimensional (3D) electro-anatomical mapping system and intracardiac echocardiography, allows to reduce the fluoroscopy time and thus the fluoroscopy exposure [82]. The use of this technique, however, requires a long learning curve [83] by the staff and it also characterized by longer procedural times than cryoablation.

Furthermore, compared to RF ablation, CB ablation produces clinical outcomes that are more reproducible and less operator-dependent [84]. CB ablation may therefore be especially advantageous for centres with less seasoned operators and fewer CAs per year. Another benefit of CB-based PVI for facilities that do not do AF ablation under severe sedation is the reduced operation time [81]. For all these reasons, the choice should be personalized on the patient and appropriate to the expertise of the center.

## 7. Conclusions

Given the extensive overlap and intertwining of clinical manifestations, risk factors, and consequences that characterise both of these diseased entities, the management of atrial fibrillation in cancer patients is a particularly challenging issue. The appropriate therapy to be given to cancer patients with atrial fibrillation has only recently attracted a little amount of attention. Studies assessing the potential trialability and safety of these therapies in cancer patients with AF are limited and, particularly with regard to LAAC and PVI, partially inconclusive. Future research will be required to thoroughly examine how cancer patients with AF respond to these different treatment regimens, with particular emphasis on the substantial haemorrhagic and thrombotic risk these patients experience.

## Figures and Tables

**Figure 1 jcm-11-04452-f001:**
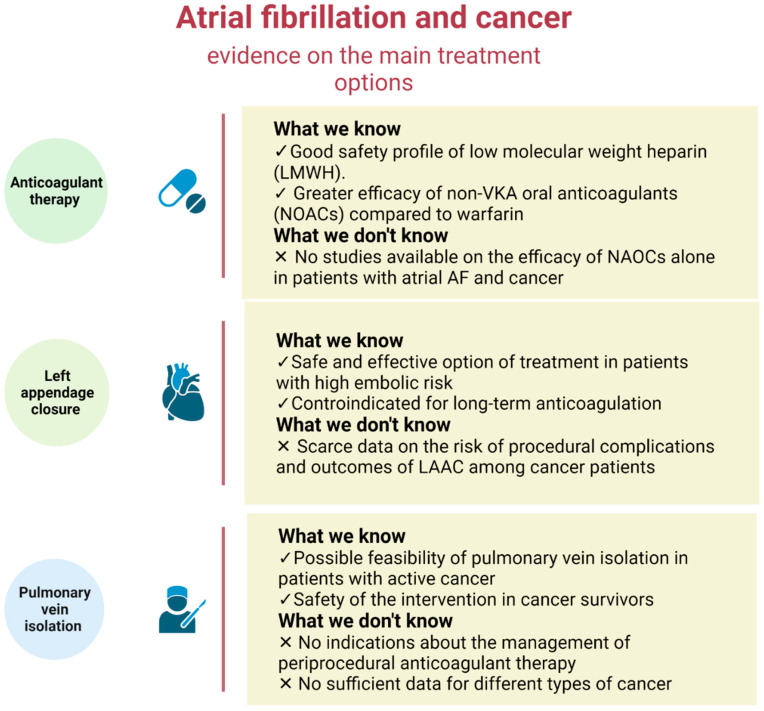
Overview of possible therapeutic strategies for atrial fibrillation in cancer patients. The main therapeutic strategies used to treat atrial fibrillation have only been partially explored in the cancer patient. Although a number of studies have advanced our knowledge of how to manage the oncological patient with atrial fibrillation, there are still a fair number of questions that have not yet been fully answered. Created with Biorender.com (accessed on 13 July 2022).

**Table 1 jcm-11-04452-t001:** Efficacy and Safety outcomes of NAOCs in AF cancer Patients.

Reference	Study Design	Mean Age (year)	Male Sex (%)	AF Patients with History of Cancer and/or Active Cancer (%)	Type of Cancer	Efficacy Outcomes	Safety Outcomes	NOAC	Follow Up (Year)
Chen et al. (2019) [14]	Post-hoc analysis from ROCKET AF trial	77	423 (66%)	640 (%4.5)	Prostate (28.6%), breast (14.7%), colorectal (16.1%), gastrointestinal (3%), lung (3.1%), melanoma (5.9%), leukemia or lymphoma (5.2%), gynecological (6.6%), genitourinary (12.2%), head and neck (3.9%), thyroid (2.5%), brain (0.3%), others (3%), unspecified cancer type (3.9%)	Stroke or systemic embolism, ischemic stroke, hemorrhagic stroke, myocardial infarction, venous thromboembolism, all-cause death, cardiovascular death	Major bleeding *, intracranial bleeding, non-major clinically relevant (NMCR) bleeding, any bleeding	Rivaroxaban	1.9
Shah et al. (2018) [15]	Retrospective population-based cohort study	74	2430 (40%)	6075 (100%)	Breast (19.2%), gastrointestinal (12.7%), lung (12.3%), genitourinary (29.2%), gyneco-oncological (2.4%), hematological (9.8%), others (14.4%)	Ischemic stroke, venous thromboembolism	Severe bleeding (intracranial or gastrointestinal), other bleeding	Rivaroxaban, dabigatran, apixaban	1.0
Fanola et al. (2018) [16]	Post-hoc analysis from ENGAGE AF-TIMI 48 trial	75	794 (68.9%)	1153 (5.5%)	Prostate (13.7%), breast (6.5%), bladder (7.5%), gastrointestinal (20.5%), lung or pleura (11%), skin (5.9%), pancreatic (3.8%), liver, gallbladder, or bile ducts (3.8%), esophageal (2.5%), oropharyngeal (2.6%), renal (2.5%), uterine (2.1%), brain (2.1%), genital (1.3%), thyroid (1.1%), leukemia (2.8%), lymphoma (2.2%), others (1.3%), unspecified cancer type (1.5%)	Stroke or systemic embolism, ischemic stroke, myocardial infarction, all-cause death, cardiovascular death	Major bleeding *, gastrointestinal bleeding, NMCR bleeding, any bleeding	Edoxaban	2.8
Kim et al. (2018) [17]	Retrospective population-based cohort study	72.4	267 (68.8%)	388 (100%)	Stomach (20.6%), colorectal (14.9%), thyroid (10.8%), prostate (9.3%), lung (12.2%), melanoma (5.9%), biliary tract (5.4%), urinary tract (6.1%), genitourinary (12.2%), head and neck (4.1%), hepatocellular carcinoma (3.0%), breast (2.4%), ovary and endometrial (2.6%), renal cell carcinoma (3.1%), hematologic malignancy (2.2%), others (3.2%)	Stroke or systemic embolism, ischemic stroke, all-cause death	Major bleeding *, gastrointestinal bleeding, intracranial bleeding, other bleeding	Rivaroxaban, dabigatran, apixaban	1.8
Melloni et al. (2017) [18]	Post-hoc analysis from ARISTOTLE trial	75	831 (67.2%)	1236 (6.8%)	Bladder (7%), breast (16%), colon (11%), gastric (2%), lung (3%), melanoma (6%), others (10%), ovarian/uterus (6%), prostate (29%), rectal (3%), renal cell carcinoma (4%), Hodgkin’s lymphoma (1%), leukemia (<1%), lymphoma (1%), non-Hodgkin’s lymphoma (1%)	Stroke or systemic embolism, myocardial infarction, all-cause death	Major bleeding *, NMCR bleeding, any bleeding	Apixaban	1.8
Ording et al. 2017 [19]	Retrospective population-based cohort study	<65 (168) 65–74 (580)75–79 (336)≥80 (725)	886 (49%)	1809 (15.2%)	Urological (15%), breast cancer (12%), GI (12%), lung (4%), hematological (3%), intracranial (0.1%), other sites (54%)	Recurrence of ischemic stroke, VTE, other arterial embolism, or myocardial infarction	Diagnosis of hemorrhagic stroke or GI, lung, or urinary hemorrhage	Not referred	1
Flack et al. 2017 [20]	Post-hoc analysis from RE-LY trial	76.4	22 (64.7%)	34 (77.2%)	Not specified		Major bleeding * due to a GI cancer	Dabigatran	2.2
Laube et al. 2017 [22]	Retrospective cohort study, single center	72	92 (56%)	163 (100%)	Lung (19%), hematologic (15%), GI (12%), genitourinary (11%), breast (10%), other (33%)	Stroke, systemic embolism	Death, CRNMB leading to discontinuation of the drug for at least 7 d Major bleeding *	Rivaroxaban	2
Russo et al. 2018 [23]	Retrospective cohort study, single center	73.2	48 (63%)	76 (100%)	Prostatic (22%), breast, (18%), colorectal (15%), gastric (3%), lung (8%), bladder (8%), kidney (4%), esophageal (3%), skin (4%), laryngeal (3%)	Ischemic stroke TIAd, Systemic embolism	Major bleeding *. All other bleedings were classified as minor	Dabigatran (37) Apixaban (21) Rivaroxaban (18)	4
Ianotto et al. 2017 [24]	Case–control study	68.6	6 (46%)	13 (1.7%)	Myeloproliferative neoplasm;	Any documented thrombosis	Major bleeding *. All other bleedings were classified as minor	Rivaroxaban (6) Apixaban (6) Switch from apixaban to rivaroxaban (1)	2.1

NMCR, non-major clinically relevant bleeding; * (ISTH criteria).

**Table 2 jcm-11-04452-t002:** PVI in cancer patients.

Study (year)	Mean Age (year)	Male *n* (%)	History of Cancer/Active Cancer *n*	Type of Tumor (%)	Type of Arrhythmia *n* (%)	Procedure Length (min)	Adverse Event (*n*)	Restoration of Sinus Rhythm after Ablation *n*	Follow Up (Year)
Kanmanthareddy et al. (2015) [63]	63 ± 7	10 (100)	10	Not specified	Atrial fibrillation (AF) 15 (100)	200 ± 33	Groin hematoma (2)	12	1
Shabtaie et al. (2021) [64]	62.4 ± 9.3	9 (53)	17	Neuroendocrine tumors (100)	Atrial flutter 2 (11.7) AF 4 (23.5) Atrioventricular nodal reentrant tachycardia 7 (41.2) Premature ventricular contractions 3 (17.6) Ventricular Tachycardia 1 (5.9)	196.4 ± 108.5	Deep venous thrombosis (1) Cardiac tamponade (1) access site bleeding A1)	2	1.6 ± 2.2
Giustozzi et al. (2021) [65]	64.3 ± 7.5	14 (67)	21	Solid tumors (95.2) Haematologic tumor (4.8)	AF 21 (100)	Not specified	Clinically relevant bleedings (4) Peri-procedural thromboembolic event (1)	13	0.08 ± 0.013
Eitel et al. (2021) [66]	71.3 ± 8.3	39 (55.7)	70	Genitourinary cancer (30), breast cancer (28.6), haemato-oncologic cancer (12.9), gastrointestinal cancer (11.4), head or neck cancer (5.7), lung cancer (2.9)	AF 7 (100),	128.7 ± 36.1	Phrenic nerve palsy (4) Pseudoaneurysm (2)	47 *	1.68 ± 0.97
Ganatra et al. (2020) [67]	65.5	81 (50)	162	Breast cancer (30.8) Other types of cancer (69.2)	AF 162 (100)	Not specified	Access site bleeding (5) Non-access site bleeding (4) strokes (2) Cardiac tamponade (2) pulmonary vein stenosis (1)	133 *	Not specified

AF: Atrial Fibrillation. * Referred to the first year of follow up.

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
