# Peer review of "Treatment Options in AF Patients with Cancer; Focus on Catheter Ablation"

_jcm, 2022, doi:10.3390/jcm11154452_

Round 1

Reviewer 1 Report

The authors present a review article titled "Catheter Ablation: Safety and Feasibility in the Oncologic Patient with Atrial Fibrillation" Congratulations to the authors for completing this review and presenting this manuscript. The manuscript is well written. Title is a bit misleading. Although catheter ablation is discussed well in the manuscript, other aspects of management of atrial fibrillation including anticoagulation and LAAO devices have been discussed as well. I suggest the authors consider re-titling the manuscript to show a more general review. Overall, only minor spelling errors- heading 4 and 5- multiple areas “AF” is listed as “FA”.

Author Response

  • “.. Title is a bit misleading. Although catheter ablation is discussed well in the manuscript, other aspects of management of atrial fibrillation including anticoagulation and LAAO devices have been discussed as well. I suggest the authors consider re-titling the manuscript to show a more general review.”

We thank the reviewer for this insightful suggestion. We agree with the choice of a more generic title to fully represent the entire content of the review. As a result, we formulated a title that better fits the manuscript's content.

  • “… Overall, only minor spelling errors- heading 4 and 5- multiple areas “AF” is listed as “FA”.”

All spelling errors involving the word "FA" have been corrected.

Reviewer 2 Report

Dear Authors,

Congratulations on your very well prepared review work. 

I have two comments: 

1. The latest 2020 ESC Gudelines Atrial Fibrillation should be included in bibliography

2. You should also refer to the document: Steffel J., Collins R., Antz M. 2021 European Heart Rhythm Association Practical Guide on the Use of Non - Vitamin K Antagonist Oral Anticoagulants in Patients with Atrial Fibrillation. Europace 2021 Oct 9;23(10):1612-1676 doi: 10.1093/europace/euab065.

Best regards

Author Response

  1. The latest 2020 ESC Gudelines Atrial Fibrillation should be included in bibliography
  2. You should also refer to the document: Steffel J., Collins R., Antz M. 2021 European Heart Rhythm Association Practical Guide on the Use of Non - Vitamin K Antagonist Oral Anticoagulants in Patients with Atrial Fibrillation. Europace 2021 Oct 9;23(10):1612-1676doi: 10.1093/europace/euab065.

We thank the reviewer for the insightful suggestions. The suggested bibliographical references have been added to the bibliography.

Reviewer 3 Report

The subject is very interesting. The paper is a narrative review with updated informations about oncological patients. The information is presented like in a book chapter. However, the article lacks conclusions. I would also suggest  the addition of a vignette containing a briefing of what is known about the subject.

Author Response

  1. “… However, the article lacks conclusions.”

We thank the reviewer for the comment. we have inserted a new paragraph concerning the conclusions

  1. “… I would also suggest  the addition of a vignette containing a briefing of what is known about the subject.”

We thank the reviewer for the insightful suggestion.

On this regard, in keeping with the literature on the treatment of cancer patients who have atrial fibrillation, we have added an explanation graphic.